# Sexual reproduction of the snow alga *Chloromonas fukushimae* (Volvocales, Chlorophyceae) induced using cultured materials

Ryo Matsuzaki[1,2]*, Masanobu Kawachi[2], Hisayoshi Nozaki[3], Seiichi Nohara[2], Iwane Suzuki[1]

1 Faculty of Life and Environmental Sciences, University of Tsukuba, Tsukuba, Ibaraki, Japan, 2 Center for Environmental Biology and Ecosystem Studies, National Institute for Environmental Studies, Tsukuba, Ibaraki, Japan, 3 Department of Biological Sciences, Graduate School of Sciences, University of Tokyo, Bunkyo-ku, Tokyo, Japan

* matsuzaki.ryo@nies.go.jp

**Data Availability Statement:** All relevant data are within the manuscript and its Supporting Information files.

## Abstract

Snow algae are microalgae, growing in melting snowpacks, and are thought to act as primary producers in the snow ecosystem. *Chloromonas* (Volvocales, Chlorophyceae) contains more than 15 snow-inhabiting species. Although vegetative cells and zygotes, or asexual cysts, of snow species of the genus are frequently collected in the field, sexual reproduction and zygote formation in culture have only been induced in *C. tughillensis*. Here we describe the sexual reproduction of another snow-inhabiting species, *C. fukushimae*, which was induced using both previously examined and newly established Japanese strains. Mating of isogamous gametes began after mixing two different strains, implying that *C. fukushimae* is an outcrossing species. Motile and nonmotile zygotes of the species were also described in this report. The nonmotile zygote of *C. fukushimae* was distinguishable from those of the other snow-inhabiting species of *Chloromonas*, based on the zygote shape and the presence of several large lipid bodies within the cell. In addition, *C. fukushimae* carried out sexual reproduction and produced zygotes even under the nitrogen-sufficient condition.

## Introduction

Snow algae, microalgae growing in melting snowpacks, are thought to act as primary producers in the snow ecosystem [1,2]. *Chloromonas* (Volvocales, Chlorophyceae), a unicellular biflagellate genus, includes more than 15 snow-inhabiting species in addition to approximately 130 mesophilic species [2,3]. Before the snowpack melts completely, the snow-inhabiting species form zygotes or asexual cysts that later enter a dormant state [1]. Mature zygotes and cysts accumulate orange carotenoid pigments within the cell, due to a protective reaction against excessive visible and ultraviolet light. The blooms of snow-inhabiting *Chloromonas* species

**Funding:** RM was supported by Grant-in-Aid for JSPS Research Fellow (No. 16J09828) from the Ministry of Education, Culture, Sports, Science and Technology (MEXT)/Japan Society for the Promotion of Science (JSPS) KAKENHI, Japan (https://www.jsps.go.jp/english/e-pd/index.html) and the grant from the Institute for Fermentation, Osaka (IFO), Japan (No. Y-2019-008) (http://www.ifo.or.jp/research/guide05.html). The funders had no role in study design, data collection and analysis, decision to publish, or preparation of the manuscript.

**Competing interests:** The authors have declared that no competing interests exist.

color snow green if vegetative cells, planozygotes (motile zygotes), or immature cysts or aplanozygotes (nonmotile zygotes) dominate. The snow is stained orange, brown or reddish if mature aplanozygotes and/or cysts are dominant within the blooms. In Antarctica, red snow caused by vegetative cells of *C. rubroleosa*, which contain amount of red-pigmented oil droplets within the cell, was also reported [4]. These phenomena are called "colored snow" (S1 Fig).

To date, more than one hundred strains of *Chloromonas* originating from snow are available from public culture collections [e.g., Culture Collection of Algae at the University of Texas at Austin [5], Culture Collection of Cryophilic Algae [6], and Microbial Culture Collection at the National Institute for Environmental Studies (NIES) [7]]. Recently, several snow-inhabiting species of *Chloromonas*, such as *C. fukushimae*, were described based on a combination of comparative morphological analysis and molecular data of cultured materials (i.e., polyphasic approaches) [8–11]. However, for most species, there is insufficient information about their aplanozygotes because sexual reproduction (including aplanozygote formation) in culture was only accomplished in *C. tughillensis* [12]. *Chloromonas tughillensis* is an outcrossing species. Its sexual reproduction was induced mixing different mating types (mt+ and mt−) that were maintained under nitrogen-depletion conditions for almost three weeks and placed under blue light during the final seven days of growth [12]. Nitrogen-depletion and blue light are thought to be classic gamete formation triggers in freshwater green algae [13,14]. The method also induced the sexual reproduction of a self-fertilizing snow species, *C. chenangoensis* [12]. Despite these successful inductions, aplanozygotes were not observed in the experiment. On the other hand, recent molecular data identified the aplanozygotes, or asexual cysts, of the four snow-inhabiting species (*C. hindakii*, *C. miwae*, *C. muramotoi*, and *C. krienitzii*) from field-collected materials [9,11,15].

Here, we induced the sexual reproduction of *C. fukushimae*, using both previously examined and newly established strains. The mating of gametes and the morphologies of planozygote and aplanozygote in this species were described, and aplanozygote morphologies were compared between *C. fukushimae* and the other snow-inhabiting species of *Chloromonas*. Additionally, we compared the induction rates of the gametes and the planozygotes in the sexual reproduction of *C. fukushimae* under nitrogen-sufficient and -deficient conditions.

## Materials and methods

### Ethics statement

We collected colored snow from snowpacks in Mt. Hakkoda in Towada–Hachimantai National Park and in Oze National Park. No specific permission was required because collecting snowpacks containing microalgae or other protists from national parks is not legally restricted in Japan. In addition, we confirmed the samples did not contain protected organisms.

### Establishment of new strains

For this study, seven new strains of *C. fukushimae* were established using the pipette-washing method [16]. Six of the seven strains (HkCl-106, HkCl-108, HkCl-113, HkCl-117, HkCl-121 and HkCl-125) were isolated from brown snow material, collected on Mt. Hakkoda, Aomori, Japan (40˚38'49.50" N, 140˚51'04.70" E) on May 18, 2016 (S1A Fig). The remaining strain (OzCl-11) originated from a green snow sample from Oze National Park, Gunma, Japan (36˚ 54'32.91" N, 139˚11'50.81" E) on April 29, 2017 (S1B Fig). The new strains were identified based on their vegetative morphology. The new strains HkCl-117 and OzCl-11 were deposited in NIES [7] as NIES-4452 and NIES-4453, respectively. Two previously examined strains of *C.*

*fukushimae* (NIES-3389 and NIES-3390) [8] were also used to induce sexual reproduction. The cultures were grown in screw-cap tubes (18 × 150 mm) containing 10 mL of artificial freshwater-6 (AF-6) medium ([17], modified according to [7]) at 5˚C, with a light:dark cycle of 14:10 h under cool-white light-emitting diodes (color temperature = 5000 K) at 35–90 μmol $m^{-2} s^{-1}$. The conditions were led from those in the fields from which the colored snow materials were collected.

## Induction of sexual reproduction

Sexual reproduction was induced according to the method for a mesophilic green alga *Chlamydomonas reinhardtii* (Volvocales, Chlorophyceae) [18], with some modifications. For comparison, the following experiments were performed using AF-6 medium (including 22 mg $NH_4NO_3 L^{-1}$ and 140 mg $NaNO_3 L^{-1}$) [7,17], and AFM medium [19] containing very little nitrogen (2.2 mg $NaNO_3 L^{-1}$), respectively. Each seven-day-old culture in 10 mL of AF-6 medium was centrifuged at 5˚C and concentrated to approximately 0.2 mL. The culture was washed with 4 mL of AF-6 or AFM medium on ice. After re-concentration, each culture's cell density was equalized at $1.5 \times 10^6$ cells $mL^{-1}$; this emulated maximum population density in field-collected samples [1,12]. A pair of 0.5 mL of each culture was mixed in a well of Cellstar 24-well culture plate (Greiner Bio-One, Kremsmünster, Austria) in a cold experimental room at 5˚C and maintained as described above. For continuous observation, mating experiments were carried out just after the dark period.

To assign mating types (mt+ and mt−) of the examined strains, we conducted the tunicamycin sensitivity test [20]. Before induction of sexual reproduction, cells in each culture were treated by 2.5 μg $mL^{-1}$ tunicamycin (Sigma-Aldrich, Missouri, USA) for overnight. Sexual reproduction of treated and untreated (positive control) cultures were induced as described above. Based on the previous study [20], a tunicamycin-sensitive mating type was assigned to mt+.

## Morphological observation and cell counting

Phase contrast microscopy of living cells was carried out in the cold experimental room, using a Zeiss Primovert inverted light microscope (Zeiss, Oberkochen, Germany) and a ScopePad-550 digital camera (Ouyan International, Shenzhen, China). Cells were fixed for counting, using 0.4% of Lugol's iodine solution, at one, two, three, four, six, eight, 10, and 12 hours after mixing the strains in the experiments started just after the dark period. Standard glutaraldehyde solutions were not used in this study since they cause the loss of flagella from cells during fixation. Using an Olympus CKX41 microscope (Olympus, Tokyo, Japan) at room temperature, the fixed cells were categorized as vegetative cells or gametes, autosporangia with daughter cells, mating pairs, and quadriflagellate planozygotes. In respective times, 400-cell counts were conducted according to the previous study [21]. The proportion of cell types among 100 mating pairs was also calculated at each time. We compared the induction rates and the proportion of cell types under the nitrogen-sufficient condition (AF-6) with those under the nitrogen-deficient condition (AFM), using Welch's *t* test. $P < 0.01$ was considered significant. In addition, light and fluorescent microscopy of fixed gametes and planozygotes, and living aplanozygotes were also performed, at room temperature, using an Olympus BX51 light microscope equipped with Nomarski differential interference optics and an Olympus DP72 digital camera (Olympus). To detect lipid bodies within the cell, the aplanozygotes were stained by 1 μg $mL^{-1}$ Nile Red for more than 48 hours at 5˚C.

## Results

### Sexual reproduction of *Chloromonas fukushimae*

The strains of *C. fukushimae* used in this study (excluding NIES-3389) showed an outcrossing feature (i.e., heterothallism) and were classified into two complementary mating types. One contained HkCl-106, HkCl-108, HkCl-113, HkCl-117, HkCl-121 and HkCl-125, and the other included OzCl-11 and NIES-3390 (S1 Table). The remaining strain, NIES-3389, did not cross with the other strains examined. Since mating type-specific genes such as *MID* in *Chlamydomonas reinhardtii* [22] have not been reported in the genus *Chloromonas*, we conducted the tunicamycin sensitivity test [20] to assign mt+ and mt−. Unilateral effect of inhibiting gamete interactions by tunicamycin was observed in some outcrossing species of *Chlamydomonas* [20] and snow-inhabiting *Chloromonas tughillensis* [12]. The test using the representative strains of the complementary mating types exhibited that the strain HkCl-117 was more sensitive to the inhibition of aplanozygote formation by tunicamycin treatment than the strain NIES-3390 (Table 1). Based on the previous results [20], the mating type including the strain HkCl-117 was assigned to mt+ in this study.

In the sexual reproduction of *C. fukushimae*, pairing isogamous gametes with entwined flagella (Fig 1A) began two to three hours after mixing mt+ and mt− strains. The gametes were elongate-bean-shaped and morphologically resembled the vegetative cells [8]. The gametes measured 4.0–6.5 μm wide and 12.8–16.1 μm long. Each gamete lacked the eyespot. Autosporangium formation was not observed from the mixing of mt+ and mt− strains to the beginning of gamete pairing. Polyvalent gamete interactions (i.e., agglutination or clumps) was occasionally found during the experiment (Fig 1B), as in the sexual reproduction of some volvocalean green algae [23] such as mesophilic *Chlamydomonas reinhardtii* [14] and snow-inhabiting *Chloromonas tughillensis* [12]. During the sexual reproduction process, one of the gametes released its cell wall and became spherical (Fig 1C and 1D; S2 Fig). Then, the other gamete also cast off the cell wall and became spherical (Fig 1E and 1F). Such morphological change of gametes was also reported in snow species of *Chloromonas* [12] as well as in several *Chlamydomonas* species [3,14,24]. Mating papillae, or fertilization tubules, were not recognized in the gametes. Finally, the pair of two spherical gametes fused and formed a quadriflagellate planozygote (Fig 1G and 1H). The planozygotes were broad ellipsoidal or almost spherical, lacked an eyespot, and measured 9.5–12.5 μm wide and 10.6–15.7 μm long. Within a few days, the planozygotes lost their flagella and became aplanozygotes (Fig 1I–1L). The aplanozygotes were ellipsoidal, broad ellipsoidal or almost spherical with the smooth cell wall, and measured 12.3–19.1 μm wide and 14.4–22.5 μm long (Fig 1I and 1K). Several large lipid bodies were observed within the cell (Fig 1J and 1L). Within two months, the aplanozygotes accumulated orange carotenoid pigments and appeared mature (Fig 1M and 1N). Neither aplanozygote meiosis nor vegetative cell germination from the aplanozygotes were observed in this study.

**Table 1. Results of the tunicamycin treatment in the sexual reproduction of *Chloromonas fukushimae*.**

| | | NIES-3390 | |
|---|---|---|---|
| | | treated | untreated |
| HkCl-117 | treated | – | – |
| | untreated | Z$^-$ | Z |

Explanation of symbols: Z, aplanozygotes were observed within 7 days; Z$^-$, aplanozygotes were observed within 7 days but the number of them was fewer than that in the untreated group; –, aplanozygotes were not observed within 7 days.

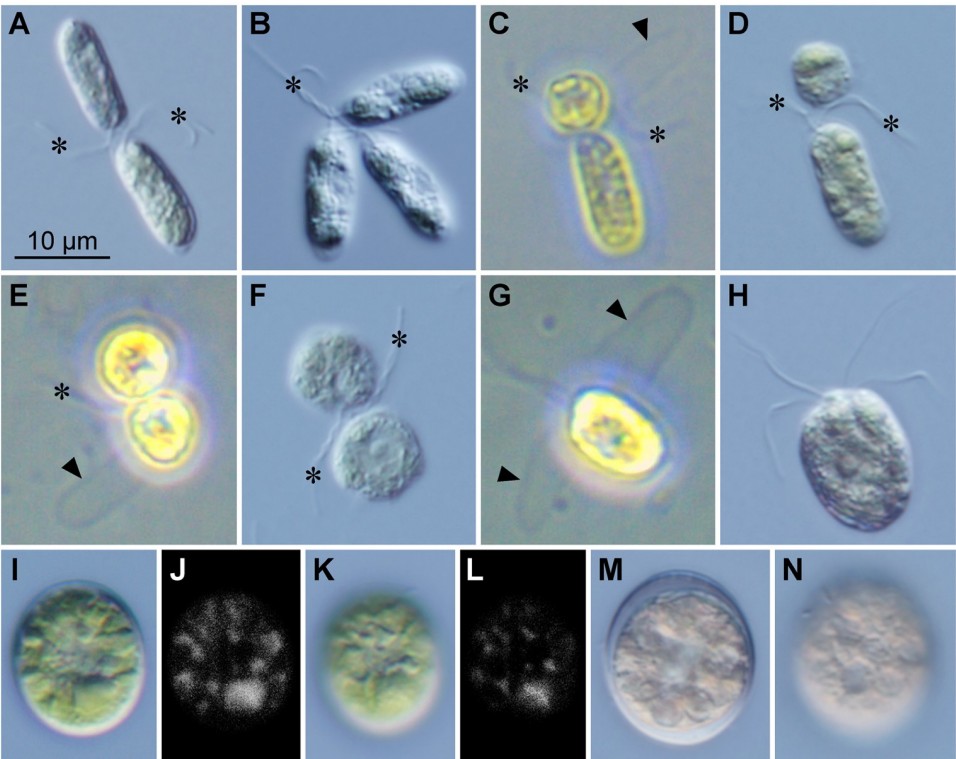

**Fig 1. Sexual reproduction of *Chloromonas fukushimae*.** Each view has identical magnification. Strains HkCl-117 (mt+) and NIES-3389 (mt−) were used. An asterisk indicates entwined flagella, and an arrowhead indicates a cell wall released from a gamete. (A) Differential interference contrast microscopy (DIC). Two elongate gametes (fixed). (B) DIC. Agglutination of three elongate gametes (fixed). (C, D) A spherical gamete and an elongate gamete. (C) Phase contrast microscopy. (D) DIC of fixed gametes. (E, F) The pairing of two spherical gametes. (E) Phase contrast micrograph. (F) DIC of fixed gametes. (G, H) Quadriflagellate planozygote resulting from the fusion of two gametes. (G) Phase contrast micrograph showing two of the four flagella. (H) DIC of a fixed planozygote. (I–L) A young aplanozygote 3 weeks after the induction of sexual reproduction, stained by Nile Red. (I) DIC of an optical section. (J) Epifluorescence image of (I). (K) DIC of a surface view. (L) Epifluorescence image of (K). (M, N) Mature aplanozygote after 3 months from the induction of sexual reproduction. (M) DIC of an optical section. (N) DIC of a surface view.

## Time scale of mating under different nitrogen conditions

Mating pairs (Fig 1A–1F) and planozygotes (Fig 1G and 1H) were observed under both nitrogen-sufficient (AF-6) and -deficient (AFM) conditions, after two to three or eight hours, respectively, from mixing mt+ and mt− strains (Fig 2A). At each time, the induction rates of mating pairs and planozygotes under the nitrogen-sufficient condition were not significantly different from those under the nitrogen-deficient condition. In addition, in each of the times, the proportion of cell types among 100 mating pairs under the nitrogen-sufficient condition did not significantly differ from that under the nitrogen-deficient condition (Fig 2B). Under both conditions, the rate of elongate gamete pairs (Fig 1A) gradually decreased from six hours after mixing strains. In contrast, the rate of spherical and elongate gamete pairs (Fig 1C and 1D) and that of two spherical gamete pairs (Fig 1E and 1F) increased from six and 10 hours, respectively, after the mix of strains.

## Discussion

The present method for inducing sexual reproduction in *C. fukushimae* is much simpler than the previous method applied to *C. tughillensis* strains [12]. Although sexual reproduction of

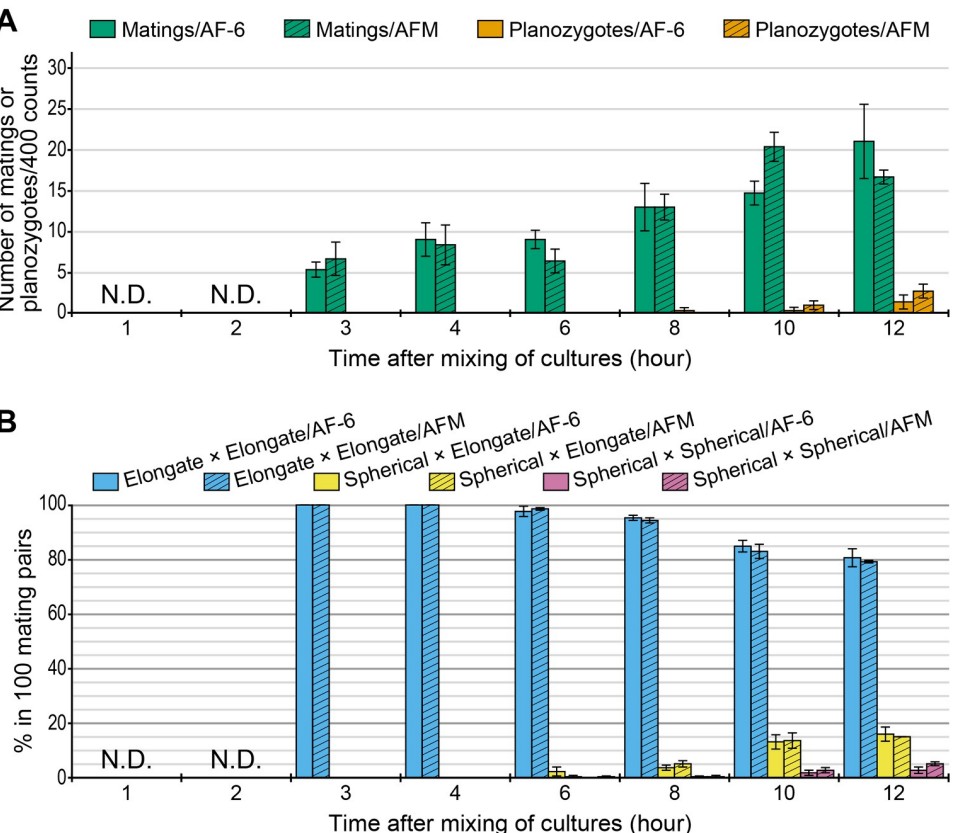

**Fig 2. Observed cell types at each time in mating experiments of *Chloromonas fukushimae*.** Strains HkCl-117 (mt+) and NIES-3390 (mt−) were used. The error bars indicate standard error ($N = 3$). N.D. means that mating pairs were not observed. (A) Observed cell types per 400 counts at each time. The numbers of vegetative cells/gametes and autosporangia with daughter cells are not shown. A pair of mating gametes was counted as one. Welch's *t* test detected no significant differences between nitrogen-sufficient (AF-6) and nitrogen-deficient (AFM) conditions throughout the experiment. (B) Observed cell types per 100 mating pairs at each time. Welch's *t* test detected no significant differences between nitrogen-sufficient (AF-6) and nitrogen-deficient (AFM) conditions throughout the experiment.

outcrossing *C. tughillensis* was previously induced by nitrogen starvation, our method was also able to induce sexual reproduction with aplanozygote formation in this species even under the nitrogen-sufficient condition (S3 Fig). Characteristics of the sexual reproduction in *C. tughillensis* observed in our experiment were consistent with those reported in the previous study [12]. Thus, this simple method is effective and may be applicable to other snow-inhabiting species of *Chloromonas*, including a self-fertilizing *C. chenangoensis*.

For induction of sexual reproduction in snow-inhabiting *C. chenangoensis* and *C. tughillensis*, optimal conditions of cell density, light intensity, photoperiod, and spectral composition were investigated [12,21,25,26]. The previous studies demonstrated that blue light and longer photoperiods significantly increased the induction rates of sexual reproduction of both species. Since the main objective of this study is to describe the sexual reproduction of *C. fukushimae*, we did not seek such optimal conditions for the species. However, the present induction rates of mating pairs and planozygotes in *C. fukushimae* (Fig 2) seems not to be much different from those in *C. tughillensis* [12].

In this study, pairing of gametes in *C. fukushimae* was observed after two to three hours from mixing of mt+ and mt− strains (Fig 2). If gametogenesis was already induced prior to the experiment or the cells were constitutively gametic, it would be expected that such the gamete

paring has occurred quickly after the mix of the strains. In the *C. tughillensis* strains maintained under nitrogen-starvation conditions for three weeks, pairing of gametes was observed after 30 minutes from the mix of mt+ and mt− strains [12]. Therefore, the time lag found in *C. fukushimae* suggests that the gametogenesis was induced by mixing with complementary mating type and/or increasing cell density. In several snow-inhabiting species of *Chloromonas*, gametes are thought to be produced via gametangium formation [12,27]. Since new gametangium formation or multiple fission was not observed from the mix of the strains to the beginning of gamete pairing, we considered that gametes of *C. fukushimae* were differentiated directly from vegetative cells after the induction. Nevertheless, it is difficult to completely rule out the possibility that a part of cells in the *C. fukushimae* strains is constitutively gametic or produces gametes via gametangium formation prior to the experiments. Besides, the gametes of this species are morphologically indistinguishable from the vegetative cells (i.e., no gamete-specific morphological characters such as mating papillae). Therefore, further studies, such as the expression analysis focusing on homologues of gamete-specific genes in *Chlamydomonas*, are required to reveal when the gametogenesis actually occurred in *C. fukushimae*.

The *C. fukushimae* strain NIES-3389 did not cross with the other strains examined in this study (S1 Table). NIES-3389 cannot be distinguished from NIES-3390 (mt−) by morphological and molecular data [8]. We hypothesize that NIES-3389 lost the ability to reproduce sexually during subculturing. However, we cannot reject the possibility that *C. fukushimae* actually contains more than one sexually isolated group (i.e., syngen).

Among the snow-inhabiting species of *Chloromonas*, the aplanozygote of *C. fukushimae* resembles that of *C. tughillensis* in that it lacks cell wall ornamentations, such as flanges or spines, and has almost spherical cell shape [12]. However, *C. fukushimae* possessed multiple well-defined lipid bodies within the cell and *C. tughillensis* does not (Fig 1I–1L) (S3F Fig; [12]). The aplanozygotes of *C. fukushimae* and *C. tughillensis*, described based on cultured materials, were never reported from field-collected materials in previous studies. Thus, future studies of snow-inhabiting *Chloromonas* should confirm that experimentally cultured aplanozygotes are morphologically identical to those formed in natural environments.

As in some freshwater green algae, nitrogen-depletion is thought to be related to shifting phases in the life cycle of snow-inhabiting *Chloromonas* [1]. On the contrary, we demonstrated that the sexual reproduction of *C. fukushimae* (and *C. tughillensis*) could be induced even under nitrogen-sufficient conditions (Fig 2; S3 Fig). The induction rates of mating pairs and planozygotes, as well as the proportion of cell types among the mating pairs in the sexual reproduction of *C. fukushimae*, were similar between nitrogen-sufficient and nitrogen-deficient conditions (Fig 2). Although further research is required, bark and leaf litters on and near colored snow might provide nutrients, including nitrogen, for snow-inhabiting species of *Chloromonas*, during the snowmelt season [1,28]. Such litter is frequently observed in snowpacks in Japanese mountainous areas. Therefore, sexual reproduction, regardless of nitrogen conditions, seems advantageous for outcrossing species of snow-inhabiting *Chloromonas* in temporal snowpacks since they must produce resting zygotes quickly before the snow completely melts away.

## Supporting information

**S1 Fig. Colored snow materials examined in this study.** (A) Brown snow in Mt. Hakkoda, Aomori, Japan (40˚38'49.50" N, 140˚51'04.70" E) on May 18, 2016. (B) Green snow in Oze National Park, Gunma, Japan (36˚54'32.91" N, 139˚11'50.81" E) on April 29, 2017.
(DOCX)

**S2 Fig. Morphological change in a gamete of the mating pair in *Chloromonas fukushimae*.** Serial phase contrast micrographs, with identical magnification throughout. Brightness and contrast were adjusted by Adobe Photoshop CS5 (Adobe Inc., California, USA) for showing entwined flagella (asterisks) and a cell wall which was being released from a gamete (arrowhead). Strains HkCl-117 (mt+) and NIES-3390 (mt−) were used. The times after beginning the observations were denoted at the upper right of each photograph. The mating types of gametes were not confirmed.
(DOCX)

**S3 Fig. Sexual reproduction of *Chloromonas tughillensis*.** Differential interference contrast micrographs with identical magnification throughout. Strains UTEX SNO91 (mt+) and UTEX SNO92 (mt−) were used under nitrogen-sufficient condition (AF-6 medium). Cell fixation was performed, using 0.4% of Lugol's iodine solution. An asterisk indicates entwined flagella. (A) The pairing of two elongate gametes (fixed). (B) The pairing of a spherical gamete and an elongate gamete (fixed). Note that the spherical gamete was smaller than the elongate one. (C) The pairing of a spherical gamete and an elongate gamete (fixed). Note that the spherical gamete was larger than the elongate one. (D) The pairing of two spherical gametes (fixed). (E) Quadriflagellate planozygote (fixed). Three of the four flagella are shown. (F) Mature aplanozygote, accumulating orange pigments within the cell.
(DOCX)

**S1 Table. Results of mating experiments among strains of *Chloromonas fukushimae*.** (DOCX)

## Acknowledgments

We are grateful to Dr. Koji Yonekura (Okinawa Churashima Foundation Research Center, Japan) for his kind help during fieldwork in Mt. Hakkoda. Fieldwork in Oze National Park was conducted as part of the 4th Oze Scientific Research.

## Author Contributions

**Conceptualization:** Ryo Matsuzaki, Masanobu Kawachi, Hisayoshi Nozaki.

**Data curation:** Ryo Matsuzaki.

**Formal analysis:** Ryo Matsuzaki.

**Funding acquisition:** Ryo Matsuzaki, Masanobu Kawachi, Hisayoshi Nozaki, Iwane Suzuki.

**Investigation:** Ryo Matsuzaki, Seiichi Nohara.

**Methodology:** Ryo Matsuzaki.

**Project administration:** Ryo Matsuzaki.

**Resources:** Ryo Matsuzaki, Masanobu Kawachi, Seiichi Nohara.

**Supervision:** Masanobu Kawachi, Hisayoshi Nozaki, Iwane Suzuki.

**Validation:** Ryo Matsuzaki.

**Visualization:** Ryo Matsuzaki.

**Writing – original draft:** Ryo Matsuzaki.

**Writing – review & editing:** Ryo Matsuzaki, Masanobu Kawachi, Hisayoshi Nozaki, Seiichi Nohara, Iwane Suzuki.

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
