## [Decision Letter · Decision Letter 0]

3 Jul 2020

PONE-D-20-16498

Sexual reproduction of the snow alga Chloromonas fukushimae (Volvocales, Chlorophyceae) induced using cultured materials

PLOS ONE

Dear Dr. Matsuzaki,

Thank you for submitting your manuscript to PLOS ONE. After careful consideration, we feel that it has merit but does not fully meet PLOS ONE’s publication criteria as it currently stands. Therefore, we invite you to submit a revised version of the manuscript that addresses the points raised during the review process.

This is a very nice study. Both reviewers had some comments and suggestions in separate attachements that you may address in your revision.  I also had a few comments for you to address.

1. What is the evidence that the strains are not constitutively gametic?  Was induction necessary? It would help to know what methods were tried and failed to induce gametogenesis if the cells are not constitutively gametic.  

2.  For the nitrogen experiment, is it possible that the nitrogen was used up during the seven day culturing procedure in both the normal and low nitrogen media, and that the reintroduction of nitrogen in the washed cells was not sufficient to suppress gametogenesis or cause de-differentiation? In other words, how conclusively can you say that gametogenesis does not require nitrogen starvation?

3.  Is there some conceptual reason to separate figures. 3 and 4? They seem like data from the same experiment.

4.  In Both Figs. 3 and 4 there are mating experiments done in the light and in the dark which yielded different outcomes, but this finding was not explicitly mentioned anywhere or discussed.

5. Lines 153-155. It was stated that gametes develop directly from vegetative cells because autosporangia were not observed in mating reactions. Please clearly define for readers how you characterize the distinction between direct conversion of vegetative cells to gametes versus a pathway which requires autosporangia formation. I was not clear how you could rule out the possibility that an obligatory division/autospore formation step did occur as part of gametogenesis during the 7 day incubation of cultures prior to concentration for mating.  In addition, please also note comment 1 above about whether this species might be constitutively gametic.

6. Lines 157-8.  As noted by Reviewer Goodenough, polyvalent gamete interactions (i.e. agglutination) is a well-known phenomenon in *Chlamydomonas reinhardtii*. It is not unusual or even necessarily mysterious to see clumps of 2+ mating cells.   

7. Line 115-116.  I agree with the Reviewer Goodenough about adding a better description of the tunicamycin experiment. Please note that you can use this test as a way to assign mating type, but I am not sure you can “determine” mating type as there is no accepted definition of what it means to be *plus* or *minus* across different algal taxa.

We look forward to receiving your revised manuscript.

Kind regards,

James G. Umen, Ph. D.

Academic Editor

PLOS ONE

Journal Requirements:

Reviewers' comments:

Reviewer's Responses to Questions

**Comments to the Author**

1. Is the manuscript technically sound, and do the data support the conclusions?

Reviewer #1: Yes

Reviewer #2: Yes

2. Has the statistical analysis been performed appropriately and rigorously? 

Reviewer #1: I Don't Know

Reviewer #2: N/A

3. Have the authors made all data underlying the findings in their manuscript fully available?

Reviewer #1: Yes

Reviewer #2: Yes

4. Is the manuscript presented in an intelligible fashion and written in standard English?

Reviewer #1: Yes

Reviewer #2: Yes

5. Review Comments to the Author

Reviewer #1: Review Comments to the Author

Please use the space provided to explain your answers to the questions above. You may also include additional comments for the author, including concerns about dual publication, research ethics, or publication ethics. (Please upload your review as an attachment if it exceeds 20,000 characters) (Limit 200 to 20000 Characters)

A well-performed study that adds a new sexual algal system for future analysis. A few comments are included in the ms as track changes, mostly requesting additional information

Reviewer #2: My comments are in the attachment! I do not understand why I have to repeat here what is in my attachment and therefore you should find what you need in that document. This review process is far more cumbersome than it needs to be. I have now wasted almost an hour trying to send this review back.

6. PLOS authors have the option to publish the peer review history of their article (what does this mean?). If published, this will include your full peer review and any attached files.

Reviewer #1: **Yes: **Ursula Goodenough

Reviewer #2: **Yes: **Ronald Hoham

---

## [Author Response · Author response to Decision Letter 0]

7 Aug 2020

PONE-D-20-16498

Sexual reproduction of the snow alga Chloromonas fukushimae (Volvocales, Chlorophyceae) induced using cultured materials

PLOS ONE

Dear Dr. Matsuzaki,

Thank you for submitting your manuscript to PLOS ONE. After careful consideration, we feel that it has merit but does not fully meet PLOS ONE’s publication criteria as it currently stands. Therefore, we invite you to submit a revised version of the manuscript that addresses the points raised during the review process.

This is a very nice study. Both reviewers had some comments and suggestions in separate attachements that you may address in your revision. I also had a few comments for you to address.

1. What is the evidence that the strains are not constitutively gametic? Was induction necessary? It would help to know what methods were tried and failed to induce gametogenesis if the cells are not constitutively gametic. 

Response: In this study, pairing of gametes in Chloromonas fukushimae was observed after two to three hours from the mix of mt+ and mt− strains (revised Fig 2A of the revised manuscript). If the strains were constitutively gametic, it would be expected that pairing of gametes has occurred quickly after mixing of mt+ and mt− strains. Actually, in the C. tughillensis strains maintained under nitrogen-starvation conditions for three weeks, pairing of gametes was observed after 30 minutes from the mix of strains (Hoham et al. 2006, Phycologia 45: 326, fig 27D). Therefore, we thought that the time lag found in C. fukushimae suggests that the gametogenesis was induced by our induction method (increasing cell density and/or mixing with complementary mating type). However, we agree that our data are insufficient to completely rule out the possibility that the strains are constitutively gametic. Thus, we deleted the sentences which state that gametes of C. fukushimae seem to be differentiated directly from vegetative cells, from the Abstract and the Results of the revised manuscript (Lines 30–31, 181–182 of the Revised Manuscript with Track Changes). Instead of the sentences, we added a paragraph in the Discussion of the revised manuscript to discuss both possibilities that gametogenesis occurred during the experiment or the cells were constitutively gametic (Lines 273–292 of the Revised Manuscript with Track Changes). 

2. For the nitrogen experiment, is it possible that the nitrogen was used up during the seven day culturing procedure in both the normal and low nitrogen media, and that the reintroduction of nitrogen in the washed cells was not sufficient to suppress gametogenesis or cause de-differentiation? In other words, how conclusively can you say that gametogenesis does not require nitrogen starvation?

Response: For 7-day preculturing, we used AF-6 medium only (the manuscript was revised to describe the method accurately; Line 112 of the Revised Manuscript with Track Changes). Although we did not measure actual nitrogen amount in the medium before the experiment, we don’t think that the nitrogen in the medium was used up during the period. This is because that the strains of C. fukushimae can be maintained at least 6 months under the condition with good health. 

3. Is there some conceptual reason to separate figures. 3 and 4? They seem like data from the same experiment.

Response: After reconsideration, we concluded that Figs 2B and 3B in the previous manuscript, which show the results from the mating experiments done in the dark, are not necessary for this study. Therefore, we removed them and their related sentences from the revised manuscript (Lines 134–136 of the revised manuscript with Track Changes). In the revised manuscript, Fig 3A in the previous manuscript was combined with Fig 2A and renamed Fig 2B. In the revised Fig 2, we added the results of the mating experiment at one and two hours after mixing the strains to clearly show when the pairing of gametes were observed. The caption of Fig 2 and its related sentences were also revised (Lines 133, 223, 228, 234–250, 314 of the Revised Manuscript with Track Changes). 

4. In Both Figs. 3 and 4 there are mating experiments done in the light and in the dark which yielded different outcomes, but this finding was not explicitly mentioned anywhere or discussed.

Response: Light/Dark conditions might affect the sexual reproduction of Chloromonas fukushimae, but the mechanisms were not investigated in this study. After reconsideration, we concluded that Figs 2B and 3B in the previous manuscript, which show the results from the mating experiments done in the dark, are not necessary for this study. Therefore, we removed them from the revised manuscript. 

5. Lines 153-155. It was stated that gametes develop directly from vegetative cells because autosporangia were not observed in mating reactions. Please clearly define for readers how you characterize the distinction between direct conversion of vegetative cells to gametes versus a pathway which requires autosporangia formation. I was not clear how you could rule out the possibility that an obligatory division/autospore formation step did occur as part of gametogenesis during the 7 day incubation of cultures prior to concentration for mating. In addition, please also note comment 1 above about whether this species might be constitutively gametic.

Response: If the cells of the strains constitutively produce gametes, it would be expected that pairing of gametes have occurred quickly after mixing of mt+ and mt− strains. However, in our experiments, it took two to three hours for the beginning of the pairing of gametes from the mix of the strains. Therefore, we thought that the time lag indicates that gametogenesis was induced by our method (increasing cell density and/or mixing with complementary mating type). Nevertheless, we agree that our data are insufficient to completely rule out the possibility that the strains constitutively produced gametes during the 7-day incubation. Thus, we removed the sentences which state that gametes of C. fukushimae seem to be differentiated directly from vegetative cells from the Abstract and the Results of the revised manuscript (Lines 30–31, 181–182 of the Revised Manuscript with Track Changes). Instead of the sentences, we added a paragraph in the Discussion of the revised manuscript to discuss possibilities that gametes were differentiated directly from vegetative cells, gametogenesis occurred during the experiment, or the cells were constitutively gametic (Lines 273–292 of the Revised Manuscript with Track Changes). 

6. Lines 157-8. As noted by Reviewer Goodenough, polyvalent gamete interactions (i.e. agglutination) is a well-known phenomenon in Chlamydomonas reinhardtii. It is not unusual or even necessarily mysterious to see clumps of 2+ mating cells. 

Response: Based on the comments, we deleted the sentence “The significance and the subsequent … were not clarified.” from the revised manuscript (Lines 187–188 of the Revised Manuscript with Track Changes). We revised a sentence (Lines 182–187 of the Revised Manuscript with Track Changes) to mention that polyvalent gamete interactions (i.e., agglutination) observed in the sexual reproduction in C. fukushimae were similar to those in the sexual reproduction in some volvocalean green algae, such as Chlamydomonas reinhardtii. 

7. Line 115-116. I agree with the Reviewer Goodenough about adding a better description of the tunicamycin experiment. Please note that you can use this test as a way to assign mating type, but I am not sure you can “determine” mating type as there is no accepted definition of what it means to be plus or minus across different algal taxa.

Response: We described the detailed information regarding the tunicamycin sensitivity test in the Materials & Methods and the Results (Lines 120–126, 155–168 of the Revised Manuscript with Track Changes). We also transferred S2 Table of the previous manuscript showing the results of the tunicamycin sensitivity test to the main text as Table 1 (Lines 170–174, 456–457 of the Revised Manuscript with Track Changes). We completely agree with your comment that it is not sure that we can “determine” mating type by the test. Therefore, we used “assign” instead of “determine” throughout the sections of the revised manuscript (Lines 121, 126, 161, 167 of the Revised Manuscript with Track Changes). 

We look forward to receiving your revised manuscript.

Kind regards,

James G. Umen, Ph. D.

Academic Editor

PLOS ONE

Journal Requirements:

Response: We have ensured that our revised manuscript meets PLoS One’s style requirements. 

Reviewers' comments:

Reviewer's Responses to Questions

Comments to the Author

1. Is the manuscript technically sound, and do the data support the conclusions?

Reviewer #1: Yes

Reviewer #2: Yes

2. Has the statistical analysis been performed appropriately and rigorously? 

Reviewer #1: I Don't Know

Reviewer #2: N/A

3. Have the authors made all data underlying the findings in their manuscript fully available?

Reviewer #1: Yes

Reviewer #2: Yes

4. Is the manuscript presented in an intelligible fashion and written in standard English?

Reviewer #1: Yes

Reviewer #2: Yes

5. Review Comments to the Author

Reviewer #1: Review Comments to the Author

Please use the space provided to explain your answers to the questions above. You may also include additional comments for the author, including concerns about dual publication, research ethics, or publication ethics. (Please upload your review as an attachment if it exceeds 20,000 characters) (Limit 200 to 20000 Characters)

A well-performed study that adds a new sexual algal system for future analysis. A few comments are included in the ms as track changes, mostly requesting additional information

*****Followings are the comments in the attachment from the Reviewer #1*****

L.105: Please include information on these media in re NH4 and NO3 content; not given in ref. 18. Do you know that this strain grows on NO3 alone? 

Response: We added information of NH4 and NO3 contents in AF-6 medium in the revised manuscript (Line 110 of the Revised Manuscript with Track Changes). We consider that the strains of Chloromonas fukushimae can grow on NO3 alone, since they grew in C medium (Ichimura 1971; https://mcc.nies.go.jp/medium/ja/c.pdf) that contains solely NO3− as nitrogen source. 

L.115: This needs to be described/explained and not just given refs.

Response: We described the detailed information regarding the tunicamycin sensitivity test in the Materials & Methods and the Results (Lines 120–126, 155–168 of the Revised Manuscript with Track Changes). We also transferred S2 Table of the previous manuscript showing the results of the tunicamycin sensitivity test to the main text as Table 1 (Lines 170–174, 456–457 of the Revised Manuscript with Track Changes). 

L.126–127: Meaning that all fixations used Lugol? N.B. for future reference, add some EGTA to a HEPES buffer and this shouldn’t happen with glutaraldehyde.

Response: Yes, all fixations in this study were performed by Lugol’s iodine solutions. We thank your kind advice and in future study, we will try to use glutaraldehyde solution with HEPES buffer plus some EGTA for cell fixation in snow Chloromonas. 

L.158: Chlamydomonas frequently has “clumps” with more than two gametes that go on to fuse in pairs.

Response: Based on the comments, we deleted the sentence “The significance and the subsequent … were not clarified.” from the revised manuscript (Lines 187–188 of the Revised Manuscript with Track Changes). We revised a sentence (Lines 182–187 of the Revised Manuscript with Track Changes) to mention that polyvalent gamete interactions (i.e., agglutination) observed in the sexual reproduction in C. fukushimae were similar to those in the sexual reproduction in some volvocalean green algae, such as Chlamydomonas reinhardtii. 

L.336: I tried some brightness/contrast manipulations on these figures and got better images of the flagella and wall by making the plates black and white. Suggest trying this with originals.

Response: Thank you very much for your kind advise. Based on the comments, we adjusted brightness and contrast of S2 Fig and got better images of the flagella and cell wall. We described the manipulations in the caption of S2 Fig (Lines 433–438 of the Revised Manuscript with Track Changes). 

*****Above are the comments in the attachment from the Reviewer #1*****

Reviewer #2: My comments are in the attachment! I do not understand why I have to repeat here what is in my attachment and therefore you should find what you need in that document. This review process is far more cumbersome than it needs to be. I have now wasted almost an hour trying to send this review back.

*****Followings are the comments in the attachment from the Reviewer #2*****

It was of great interest to have read “Sexual reproduction of the snow alga Chloromonas fukushimae (Volvocales, Chlorophyceae) induced using cultured material” by Matsuzaki et al. submitted to PLOS ONE for potential publication. This paper is very parallel and supportive to our publications on sexual reproduction in the snow algae, Chloromonas (Cr.) tughillensis, with many similar findings and some interesting differences. We published three papers on sexual reproduction in Cr. tughillensis of which two are cited. The third paper (published as Cr. sp.-D, the former name of Cr. tughillensis) on the effects of irradiance levels and spectral composition was published in Hydro. Process. 12:1627-39 (1998), and it would add to the Discussion if the authors included information from that paper in addition to the other two papers. 

Response: Since the main objective of our study is to describe the sexual reproduction of Chloromonas fukushimae, we did not seek the optimal conditions of cell density, light intensity, photoperiod, and spectral compositions for inducing sexual reproduction of this species. Based on the comments, we added a paragraph in the Discussion section of the revised manuscript (Lines 264–272 of the Revised Manuscript with Track Changes) to mention what kinds of parameters were examined in the previous studies for optimization of inducing mating in C. chenangoensis and C. tughillensis. 

I was very encouraged to see the authors using an experimental laboratory design to answer questions about snow algae, something that needs to be done more. The information presented here should be published as so little is know about the transformation of gametes from one shape to another in any species of Chloromonas or Chlamydomonas alone snow algae. This manuscript also lends support that Chloromonas in snow may have similar mating strategies and behaviors in different species. 

I will comment by manuscript line: 

Line 46: Cr. rubroleosa causes red snow in Antarctica and you many want to add this here (Ling & Seppelt, 1993).

Response: We added the sentence which mentions red snow in Antarctica caused by the vegetative cells of C. rubroleosa, with citation of the reference (Lines 49–51 of the Revised Manuscript with Track Changes). 

Lines 99-100: Why were photoperiods of 14:10 L:D and a Cool White irradiance level of 35-90 umol used? Did these correspond to readings from the field?

Response: Yes. Although we did not directly measure the photoperiod and light intensity of the field from which C. fukushimae samples were collected, the light:dark cycle and the light intensity in this study fall within those of the field data that we collected colored snow materials. We added the explanation in the revised manuscript (Lines 103–104 of the Revised Manuscript with Track Changes). 

Lines 122 and 126: They used Lugol’s solution as a fixative; we used 4% osmium tetroxide. We both had problems with glutaraldehyde causing flagella to fall off. We also had similar problems with Lugol’s so we used osmium. 

Response: In our observation, enough cells of snow Chloromonas seemingly kept their flagella after fixation by Lugol’s solution. We also tried using 4% osmium tetroxide for cell fixation of snow Chloromonas; however, we could not find apparent differences between osmium tetroxide and Lugol’s solutions. In addition, it is difficult for us to use OsO4 frequently for cell fixation, due to its volatile and toxicity. 

Line 147: Very interesting that strain NIES-3389 did not cross with either of the other strains. Always the unexpected when working with snow algae.

Response: Since the strain NIES-3389 was used for the holotype of Chloromonas fukushimae (Matsuzaki et al. 2014, Phycologia 53: 297), this unexpected event is also important taxonomically. 

Line 159: Just like what happened in Cr. tughillensis that oblong gametes shedding their cell walls result in spherical gametes.

Response: Yes. We added the sentence in the revised manuscript (Lines 191–192 of the Revised Manuscript with Track Changes) to mention that such morphological change of gametes was reported in C. tughillensis. 

Line 169: Zygotes in both Cr. fukushimae and Cr. tughillensis produce orange carotenoids in the lab experiments.

Response: Yes, the zygotes of the two species produce and accumulate orange carotenoids within the cell in the lab experiments. 

Lines 196-200: Very interesting comparison between the two species that both produce spherical gametes from oblong gametes through time when the wall is shed from the oblong gametes. However, there is a difference between the two species. In Cr. fukushimae, the oblong gametes remain dominant through the experiments even after spherical gametes appear whereas in Cr. tughillensis, spherical gametes become dominant after they appear (Hoham et al. 2006, p. 327, Figs. 28-29). Also see Hoham et al. 2006, p. 329, where this phenomenon of change from oblong to spherical gametes appears in species of Chlamydomonas.

Response: Since induction methods are different between C. tughillensis and C. fukushimae, we did not mention the differences regarding the ratio of oblong and spherical gametes in the revised manuscript. We added the sentence in the revised manuscript (Lines 191–192 of the Revised Manuscript with Track Changes) to mention that such morphological change of gametes was reported in snow species of Chloromonas as well as in some Chlamydomonas species. 

Lines 208-9: For Cr. tughillensis, mating experiments were started just after the dark period, which is opposite of what was done for Cr. fukushimae.

Response: We conducted two types of mating experiments for C. fukushimae: one was started just after the dark period as that for C. tughillensis and the other was started just before the dark period. Light/Dark conditions seem to affect the sexual reproduction of C. fukushimae, but the mechanisms were not investigated in this study. After reconsideration, the description regarding the mating experiment done in the dark was removed from the revised manuscript (Lines 134–136 of the revised manuscript with Track Changes). In addition, Figs 2B and 3B in the previous manuscript, which show the results of the mating experiment done in the dark, were also removed from the revised manuscript. Fig 3A in the previous manuscript was combined with Fig 2A and renamed Fig 2B. In the revised Fig 2, we added the results of the mating experiment at one and two hours after mixing the strains to clearly show when the pairing of gametes were observed. The caption of Fig 2 and its related sentences were also revised (Lines 133, 223, 228, 234–250, 314 of the Revised Manuscript with Track Changes). 

Lines 244-7: Interesting that no differences occurred in experiments with or without nitrogen. We did not test this for Cr. tughillensis as we used only nitrogen depleted cultures.

Response: To state that the sexual reproduction of C. tughillensis was also induced even under nitrogen-sufficient condition, we revised the sentences in the Discussion and the caption of S3 Fig (Lines 254–260, 443–444 of the Revised Manuscript with Track Changes). 

Lines 250-4: Interesting comments about leaf litter on snow as potential sources of nutrients for snow algae as has been shown in previous lab experiments. 

Response: We cited an additional reference of the previous lab experiments (Hoham et al. 2008, Nova Hedwigia 86: 133–140) in the revised manuscript (Line 317 of the Revised Manuscript with Track Changes). 

Fig. 1: I would have found this much easier to follow if A, B, C, and D could have been the top row and D, E, F, and G the second row, etc. reading from left to right. Their Fig. 1B is very similar to what we showed for Cr. tughillensis (Hoham et al. 2006, Fig. 20). 

Response: We re-arranged Fig 1B–1N based on the comments. 

Figs. 2-3A: How were samples removed through the dark period to avoid light entering the experiments?

Response: The samples were placed within the cold experimental room without any windows, and we did not operate the samples through the dark period. However, we removed the descriptions of the experiments done in the dark period from the revised manuscript (Lines 134–136 of the revised manuscript with Track Changes). 

S3: I was very reassured that when they repeated our mating experiments for Cr. tughillensis, they got similar results including all three mating configurations (Figs. A, C, and D) and orange carotenoids in the zygotes (Fig. F).

To strengthen the Discussion, I suggest to summarize and compare both species as to what is known about them to induce mating as observed in laboratory studies. In the data presented here it does not make any difference whether nitrogen is present or absent as there were no significant differences in matings. This was not done for Cr. tughillensis. For Cr. tughillensis, it is known that blue light (Hoham at al. 1998) and longer photoperiods (Hoham et al. 2000) significantly increase matings and sexual reproduction. For both Cr. tughillensis and Cr. chenangoensis, irradiance levels used in laboratory mating experiments corresponded to what was recorded in the field and what was found in the laboratory to be optimal. Photoperiods used were what was recorded in the field at time of appearance in snow, which was not optimal for either Cr. tughillensis or Cr. chenangoensis (Hoham et al. 2000, 2006). 

Response: Since the main objective of our study is to describe the sexual reproduction of C. fukushimae, we did not seek the optimal conditions of cell density, light intensity, photoperiod, and spectral compositions for inducing sexual reproduction of this species. We added the paragraph in the Discussion section of the revised manuscript (Lines 264–272 of the Revised Manuscript with Track Changes) to mention what kinds of parameters were examined in the previous studies for optimization of inducing mating in C. chenangoensis and C. tughillensis. 

*****Above are the comments in the attachment from the Reviewer #2*****

6. PLOS authors have the option to publish the peer review history of their article (what does this mean?). If published, this will include your full peer review and any attached files.

Do you want your identity to be public for this peer review? For information about this choice, including consent withdrawal, please see our Privacy Policy.

Reviewer #1: Yes: Ursula Goodenough

Reviewer #2: Yes: Ronald Hoham

---

## [Editor Report · Decision Letter 1]

13 Aug 2020

Sexual reproduction of the snow alga Chloromonas fukushimae (Volvocales, Chlorophyceae) induced using cultured materials

PONE-D-20-16498R1

Dear Dr. Matsuzaki,

We’re pleased to inform you that your manuscript has been judged scientifically suitable for publication and will be formally accepted for publication once it meets all outstanding technical requirements.

Kind regards,

James G. Umen, Ph. D.

Academic Editor

PLOS ONE
---

## [Editor Report · Acceptance letter]

17 Aug 2020

PONE-D-20-16498R1 

Sexual reproduction of the snow alga *Chloromonas fukushimae* (Volvocales, Chlorophyceae) induced using cultured materials  

Dear Dr. Matsuzaki:

I'm pleased to inform you that your manuscript has been deemed suitable for publication in PLOS ONE. Congratulations! Your manuscript is now with our production department. 

Kind regards, 

on behalf of

Dr. James G. Umen 

Academic Editor

PLOS ONE